# Cytoplasmic glycoengineering enables biosynthesis of nanoscale glycoprotein assemblies

Hanne L.P. Tytgat [1,4,7], Chia-wei Lin [1,5,7], Mikail D. Levasseur[2], Markus B. Tomek [1],
Christoph Rutschmann[1], Jacqueline Mock [1,6], Nora Liebscher[1], Naohiro Terasaka [2], Yusuke Azuma [2],
Michael Wetter [1], Martin F. Bachmann[3], Donald Hilvert[2], Markus Aebi[1] & Timothy G. Keys [1]*

Glycosylation of proteins profoundly impacts their physical and biological properties. Yet our ability to engineer novel glycoprotein structures remains limited. Established bacterial glycoengineering platforms require secretion of the acceptor protein to the periplasmic space and preassembly of the oligosaccharide substrate as a lipid-linked precursor, limiting access to protein and glycan substrates respectively. Here, we circumvent these bottlenecks by developing a facile glycoengineering platform that operates in the bacterial cytoplasm. The Glycoli platform leverages a recently discovered site-specific polypeptide glycosyltransferase together with variable glycosyltransferase modules to synthesize defined glycans, of bacterial or mammalian origin, directly onto recombinant proteins in the *E. coli* cytoplasm. We exploit the cytoplasmic localization of this glycoengineering platform to generate a variety of multivalent glycostructures, including self-assembling nanomaterials bearing hundreds of copies of the glycan epitope. This work establishes cytoplasmic glycoengineering as a powerful platform for producing glycoprotein structures with diverse future biomedical applications.

[1] Institute of Microbiology, ETH Zurich, 8093 Zurich, Switzerland. [2] Laboratory of Organic Chemistry, ETH Zurich, 8093 Zurich, Switzerland. [3] Department of Immunology, Inselspital, University of Bern, 3010 Bern, Switzerland. [4] Present address: Laboratory of Microbiology, Wageningen University, 6708 WE Wageningen, The Netherlands. [5] Present address: Functional Genomic Center Zurich, ETH Zurich, 8057 Zurich, Switzerland. [6] Present address: Institute of Pharmaceutical Sciences, ETH Zurich, 8093 Zurich, Switzerland. [7] These authors contributed equally: Hanne L.P. Tytgat, Chia-wei Lin. *email: tim. keys@micro.biol.ethz.ch

Across the domains of life, the cell's outermost surface is decorated with a dense array of complex carbohydrates[1]. These sugars are conduits for cell adhesion and communication, and mediators of host–microbe interactions[2], making them attractive targets for therapeutic intervention. Yet our ability to produce and engineer complex carbohydrates lags far behind the other major classes of biological macromolecules.

Glycoproteins are among the most structurally diverse class of glycoconjugates. The functional repertoire and versatility of the proteome is massively expanded by the posttranslational attachment of monosaccharides, oligosaccharides, or polysaccharides to proteins[3]. In the case of therapeutic proteins, more than 70% are glycosylated, and optimizing glycosylation can considerably benefit therapeutic efficacy[4]. To facilitate glycoengineering of the current generation of therapeutic proteins—including mAbs, hormones, and blood factors—the community has made important progress in engineering the glycosylation pathways of mammalian[5], plant[6], yeast[7], and insect cells[8] towards homogenous and humanized glycosylation profiles. On the other hand, glycoprotein engineering in bacterial cells has provided access to unprecedented structures with applications as next-generation glycoconjugate vaccines and diagnostic reagents[9].

The most advanced strategies for recombinant glycoprotein synthesis in *Escherichia coli* exploit periplasmic oligosaccharyltransferase (OST)-based pathways. OSTs are integral membrane proteins with a catalytic domain facing the lumen of the periplasm, where they catalyze glycosylation of proteins using lipid-linked oligosaccharides (LLO) as donor substrates. Three factors enable periplasmic glycoengineering: (i) the promiscuity of some OSTs towards different glycans presented on the appropriate lipid[10,11], (ii) the ability to target heterologous proteins for glycosylation by introducing an appropriate sequence motif[12,13], and (iii) a functional biosynthetic pathway (native or engineered) for producing the desired LLO. Limitations of periplasmic glycoengineering include the difficulty of engineering novel LLO biosynthesis pathways[14,15], the limited substrate promiscuity of OSTs[11,16], and the requirement to secrete acceptor proteins into the periplasm for glycosylation to occur. To provide access to new areas of glycoprotein structural space, it is essential that we develop alternative routes for bacterial glycoengineering that are not dependent on LLO intermediates, and that do not operate in the periplasm.

The cytoplasm of *E. coli* is a robust and versatile compartment for recombinant protein expression. Recent studies have shown that proteins that form functional nanoscale, megadalton assemblies can be produced in the bacterial cytoplasm[17–19]. For example, the natural cage-forming protein, lumazine synthase from *Aquifex aeolicus* (AaLS) was engineered and evolved to encapsulate various guest molecules including enzymes[20,21], fluorescent proteins[22], and most recently its own RNA genome[23,24]. Such nucleocapsids are evolvable nanostructures that can be quickly adapted to acquire important properties, such as the ability to protect cargo against nucleases[24] or increased circulatory half-life in bodily fluids[25]. These self-assembling proteins have recently attracted attention as tailored vehicles for drug delivery and vaccination. Glycosylation of the nanoparticle surface holds the potential to expand their utility in these applications, giving strong impetus to the development of cytoplasmic glycoengineering pathways.

The identification of cytoplasmic protein glycosylation systems in various bacterial species, presents exciting opportunities for cytoplasmic glycoengineering[9,26]. We have chosen the asparagine (N)-glucosyltransferase of *Actinobacillus pleuropneumoniae* (ApNGT) as the basis for a cytoplasmic glycoengineering platform. The ApNGT can be actively expressed in the *E. coli* cytoplasm and catalyzes the transfer of a single β-linked glucose onto

recombinant proteins at the N-x-S/T consensus sequon[27–29]. We have shown that this short sequon can be exploited to target glycosylation of heterologous proteins, such as the superfolder green fluorescent protein (sfGFP)[30] and next-generation antibody mimetics, such as designed ankyrin-repeat proteins (DARPin)[31,32]. In this study, we demonstrate that N-linked glucose (N-Glc) can be used as a site-specific primer for the biosynthesis of diverse oligosaccharides and polysaccharides directly onto recombinant proteins in the cytoplasm. We exploit the cytoplasmic localization of these artificial glycosylation pathways to generate a variety of self-assembling glycoproteins that form icosahedral nanostructures with future applications as vaccines and drug-delivery vehicles.

## Results

**A modular protein glycoengineering toolbox**. We have previously demonstrated the site-specific synthesis of N-linked lactose onto a protein target in *E. coli*[32]. The lactose disaccharide is present at the reducing end of a variety of common mammalian glycans, including human milk oligosaccharides (HMOs) and glycosphingolipids, making it an attractive primer for glycan biosynthesis. Furthermore, numerous biosynthetic pathways have been developed for the production of free oligosaccharides in *E. coli* using a lactose primer[33–36]. Drawing on this work, we designed protein glycosylation pathways for a range of biomedically relevant glycan epitopes (Fig. 1a).

The proposed biosynthetic pathways require co-expression of up to six enzymes from different bacterial species. To efficiently generate pathway constructs in a modular and tunable manner, we chose to combine BioBricked pathway elements (e.g. one glycosyltransferase or one nucleotide–sugar biosynthesis pathway) on plasmid backbones conforming to the standard European vector architecture (SEVA)[37]. Each glycosyltransferase brick comprised a single expression cassette including a variable promoter (lacUV5, or T5), a ribosome-binding site, the relevant open-reading frame, and a T7 terminator, flanked by the BioBrick prefix and suffix (Supplementary Fig. 1)[38]. We assembled pathway constructs (up to 13.6 kb in size) for a total of nine glycan structures, none of which, to the best of our knowledge, are accessible to existing protein glycoengineering platforms.

**Cytoplasmic glycosylation of proteins**. To screen prototype pathways for glycan assembly, they were co-expressed with a model target protein, sfGFP carrying a C-terminal extension possessing a single glycosylation site (Asn-Ala-Thr; Supplementary Fig. 2)[32]. The target protein was affinity purified and analyzed by mass spectrometry (MS). Throughout the study, we observed glycosylation exclusively at the target site, Asn263, of the sfGFP construct. The yield of sfGFP also remained constant at ~30 mg L$^{-1}$ from shaker flask culture irrespective of the glycosylation pathway.

We first established the capacity to modify proteins with common mammalian oligosaccharide epitopes, including: 2′-fucosyllactose, 3′-sialyllactose, oligosialyllactose, N-acetyllactosamine (LacNAc) repeats, and the Lewis$^X$ blood group antigen. These pathways combine glycosyltransferases from lipooligosaccharide or lipopolysaccharide biosynthesis from different bacterial species (Supplementary Table 1). Our LC–MS/MS analysis of tryptic peptides from the purified pathway products confirmed site-specific synthesis of all target glycan structures onto the protein substrate (Supplementary Figs. 3 and 4). To analyze site-occupancy and glycoform distribution we performed intact protein MS of proteins modified with glucose, lactose, 2′-fucosyllactose and 3′-sialyllactose. This semi-quantitative analysis indicated that the average (mean ± s.d., $n = 3$) occupancy of the

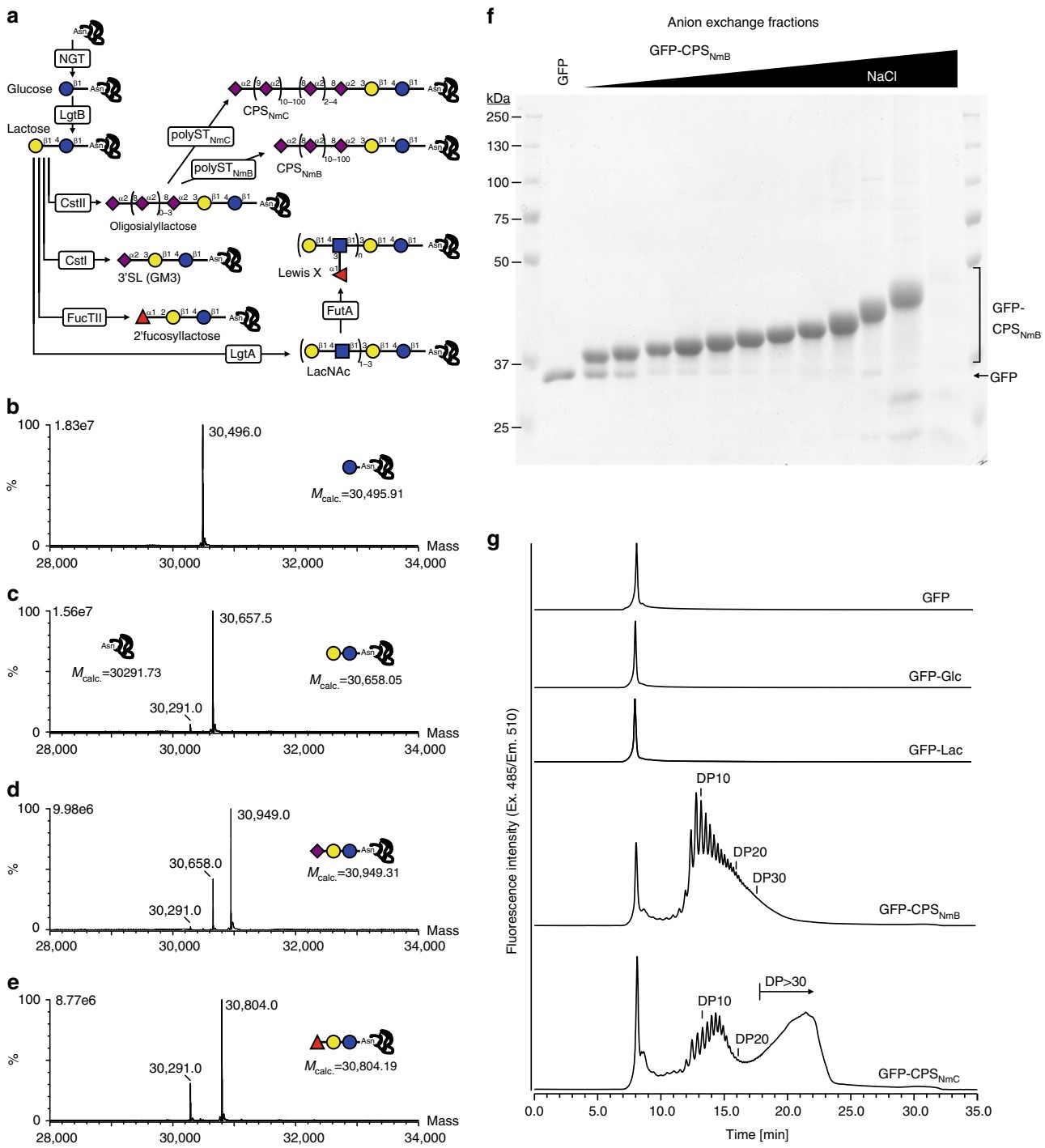

**Fig. 1** Cytoplasmic protein glycosylation. **a** Overview of protein glycosylation pathways. The protein–glycan linkage is established by transfer of a single glucose onto an asparagine residue, in the N-x-S/T sequon (where, x ≠ P), by the N-glucosyltransferase (NGT). The N-linked glucose (N-Glc) serves as a primer for combinations of glycosyltransferases from different bacterial species to synthesize a variety of oligosaccharide or polysaccharide structures on the target protein. Following co-expression with the respective glycosylation pathway, the (glyco)protein products were purified and analyzed by MS, SDS–PAGE, or analytical anion exchange chromatography. Mass spectra of intact proteins modified with **b** glucose, **c** lactose, **d** 3'-sialyllactose (3'SL), and **e** 2'-fucosyllactose. **f** Coomassie-stained SDS–PAGE of fractions from preparative anion exchange chromatography of GFP modified with CPS$_{NmB}$. **g** Purified, polysialylated GFP fractions modified with CPS$_{NmB}$ and CPS$_{NmC}$ were analyzed by anion exchange chromatography. The number of sialic acids in the polysaccharide are indicted as degree of polymerization (DP). Sugar symbols are drawn according to the guidelines of the Consortium for Functional Glycomics. Blue and yellow circles represent glucose and galactose, respectively. Red triangles and purple diamonds indicate fucose and sialic acid, respectively. Source data are provided as a Source Data file.

target sugars were 100%, 93 ± 1%, 79 ± 10%, and 62 ± 16%, respectively, with minor portions remaining unmodified or carrying a lactose intermediate (Fig. 1b–e and Supplementary Fig. 5). We used anion exchange chromatography to independently quantify the occupancy of the 3′-sialyllactose modification. We observed an average (mean ± s.d., $n = 3$) of 60 ± 8% occupancy (Supplementary Fig. 6), in excellent agreement with the MS-based analysis.

Varying pathway promoters can influence glycosylation products. The first version of each glycosylation pathway was constructed with a lacUV5 promoter driving expression of each glycosyltransferase. These constructs were effective for glucose, lactose, 2′-fucosyllactose, and 3′-sialyllactose (described above). In the case of two structures, Lewis[X] and LacNAc repeats, we only observed synthesis of the target glycan when the GlcNAc transferase, LgtA, was expressed from the strong T5 promoter. In this case, LC–MS/MS analysis indicated that the occupancy of these glycans was low (<5%). We expect that more subtle tuning of enzyme expression levels will be necessary to maximize flux through these artificial glycosylation pathways.

We next aimed to modify proteins with the capsular polysaccharides of Gram-negative bacterial pathogens, which are important targets for vaccination. Glycoconjugate vaccines—consisting of the pathogen-specific capsular polysaccharide chemically cross-linked to an antigenic carrier protein—currently confer protection against, e.g. *Haemophilus influenzae* group B and *Neisseria meningitidis* serogroups A, C, W-135, and Y[39]. To facilitate complete biosynthesis of glycoconjugate vaccines in the cytoplasm, we aimed to develop glycoengineering pathways that synthesize capsular polysaccharides directly onto protein targets.

We previously demonstrated protein-coupled biosynthesis of the nonimmunogenic polysaccharide of *N. meningitidis* serogroup B[32]. The immunogenic polysaccharide of *N. meningitidis* serogroup C is a related homopolymer of sialic acids in α2,9-linkage, and the respective capsule polymerase can be primed with a trisaccharide of sialic acid[40]. We generated pathways for both the NmB and NmC polysaccharide by combining the respective capsule polymerase with the oligosialyllactose pathway and the sfGFP acceptor. In both cases, MS analysis indicated modification of the protein with a homopolymer of sialic acid (Supplementary Fig. 7), linkage of the NmB and NmC polysaccharides was confirmed using capsule-specific antibodies (Supplementary Fig. 7). Although only a minor fraction (~15%) of total protein was modified with polysaccharide, we used preparative anion exchange chromatography to fractionate the glycoprotein products according to the size of attached polysaccharide (Fig. 1f). In this way, NmB-polysialylated and NmC-polysialylated proteins were purified with yields of 2150 and 750 µg L$^{-1}$, respectively, from shaker flask culture. The products had distinct chain-length distributions (Fig. 1g). Consistent with observations in vitro[40], the NmC-polysialyltransferase exhibited a bi-modal product distribution including a population carrying very long chains.

These results highlight the versatility of cytoplasmic glycoengineering to provide access to both small mammalian carbohydrate epitopes, as well as large polysaccharides associated with the surface of bacterial pathogens.

**Multivalent glycopolymers**. Multivalency is a common theme in glycobiology. Due to the generally low affinities of glycan–protein interactions, biological systems typically exploit multiple simultaneous interactions to achieve functionally relevant avidities[41]. In our first approach to multivalent glycan display, we introduced multiple glycosylation sites along a single polypeptide chain. Based on a consensus sequence [GANATA] for the ApNGT[27], we

designed three repeat motifs with different spacing between the glycosites: [GNAT]₅, [GANATA]₅, and [TAGANATA]₅. These sequences were genetically fused to the C-terminus of sfGFP and co-expressed with the ApNGT. Denaturing PAGE revealed a considerable increase in the apparent molecular weight of these constructs upon co-expression with the ApNGT (Fig. 2a). The modified species reacted strongly with an N-Glc-specific antiserum, MS14[28,42]. MS analysis confirmed glycosylation of the target asparagine residues and demonstrated that for all three repeat sequences, the predominant species (>80% of total) was completely glycosylated, carrying a glucose at all five sites (Supplementary Fig. 8).

To test if we could generate a uniform, multivalent glycoprotein, we increased the amount of NGT activity in the cytoplasm using an alternative ApNGT expression plasmid. Both plasmids express the ApNGT behind a lacUV5 promoter and encode chloramphenicol resistance, but differ in the presence of a pBBR1 replicon (~5 copies per chromosome) or a p15A replicon (~8.6 copies per chromosome)[43]. Each plasmid was co-expressed with the [GANATA]₅-tagged construct. Intact protein MS of the purified products revealed 93 ± 6% (mean ± s.d., $n = 3$) of the protein was modified with five glucose residues when co-expressed with the lower copy pBBR1 plasmid. In our analysis of products from the higher copy p15A plasmid, we were unable to detect species with <100% occupancy ($n = 3$), suggesting quantitative modification of the protein (Fig. 2b). These results demonstrate that improved glycosylation efficiency may be achieved with minor adjustments to enzyme expression levels.

Finally, we tested whether the closely located glucose modifications could be extended by further glycosyltransferases. The [GANATA]₅-tagged construct was co-expressed with the established pathways for lactose, 2′-fucosyllactose and 3′-sialyllactose. The LC–MS/MS data confirmed the presence of heavily glycosylated proteins carrying up to five copies of the target oligosaccharide (Fig. 2c–e and Supplementary Fig. 8). Lactose was assembled with good efficiency on all five sites, with 69 ± 18% (mean ± s.d., $n = 2$) of the protein fully modified. A much smaller portion of the protein (<5%) was completely modified with the 2′-fucosyllactose or 3′-sialyllactose at all five sites. These results indicate that a considerable increase in pathway efficiency will be necessary to achieve full occupancy of multivalent glycopolymers[44]. The ability to increase the valency of glycan display—producing structures reminiscent of mucins and lectin-blocking glycopolymers—will be critical for exploiting the biological interactions of these common oligosaccharide epitopes[44,45].

**Glycosylation of megadalton protein assemblies**. With the Glycoli platform established, we exploited its cytoplasmic localization to modify proteins that have remained inaccessible to existing periplasmic glycoengineering systems. A class of proteins that have been the center of much recent innovation are self-assembling polypeptides that form protein cages[17–19], nucleocapsids[24,25], and virus-like particles (VLPs)[46,47]. We chose to target three megadalton-scale protein assemblies for glycosylation: (i) the AP205 VLP is composed of 180 copies of the coat protein of *Acinetobacter* phage 205[48], and is a potently immunogenic scaffold for presentation of vaccine antigens[49]; (ii) AaLS-13, an engineered lumazine synthase that assembles into a six megadalton porous cage composed of 360 identical monomers and efficiently loads cargo molecules via electrostatic interactions[19,21,50]; (iii) I53-50-v4 is composed of two polypeptide sequences, which form trimeric and pentameric subunits that co-assemble into 120-mer protein assemblies[17], which have been

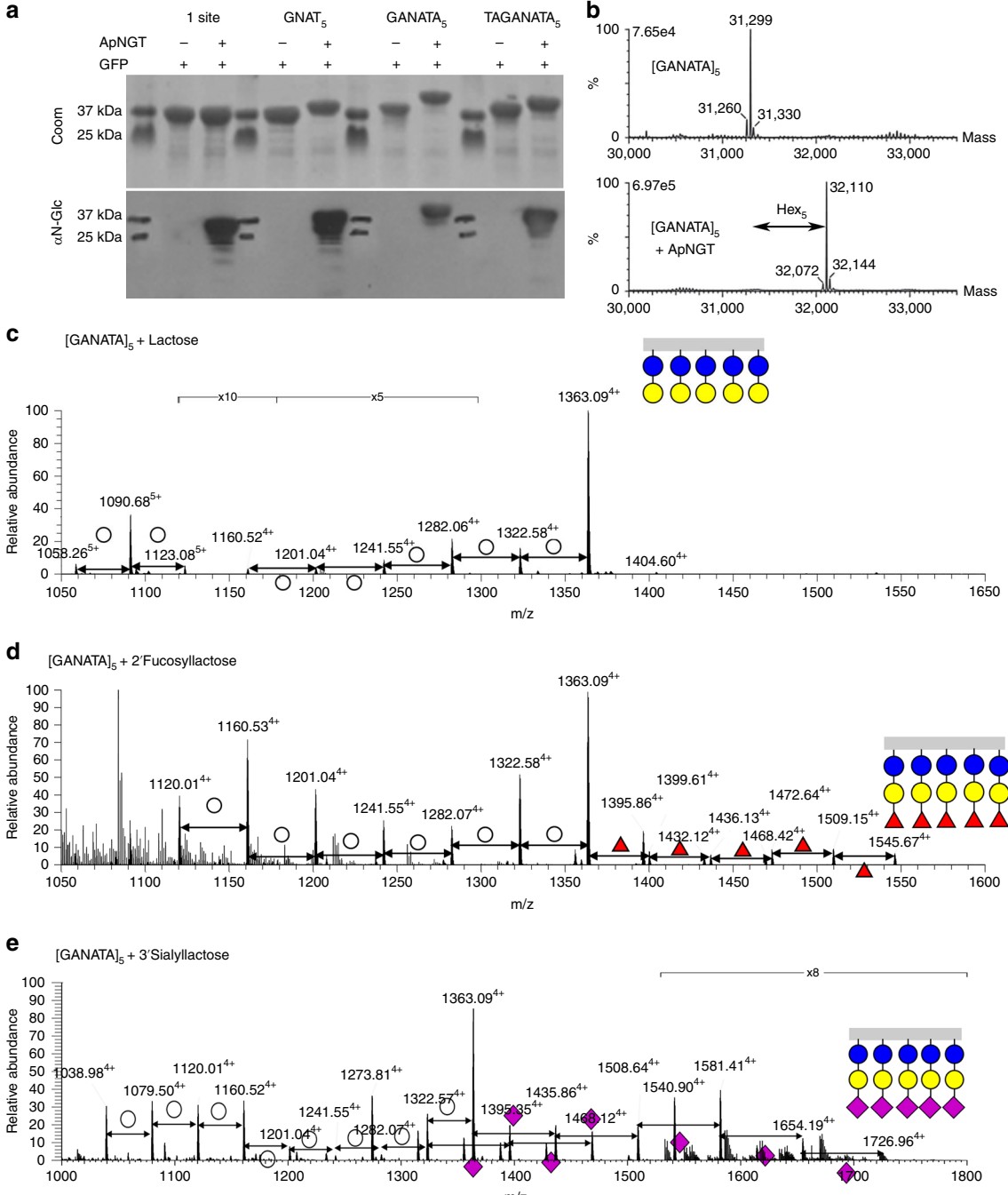

**Fig. 2** Multivalent glycopolymers. **a** Substrate proteins bearing a variety of pentavalent glycotags were co-expressed with and without the NGT, affinity purified, and analyzed by denaturing PAGE followed by Coomassie staining or immunoblot using an N-Glc-reactive serum. **b** Intact protein mass spectra of GFP-[GANATA]$_5$ expressed alone (upper panel) or with ApNGT from a p15A replicon plasmid (lower panel), showing uniform modification with five N-linked glucose residues. **c-e** Peptide LC–MS spectra show the glycosylation profile of the GFP-[GANATA]$_5$ substrate protein following co-expression with the established pathways for **c** lactose, **d** 2′-fucosyllactose, and **e** 3′-sialyllactose. The assigned structures are supported by HCD–MS/MS spectra (Supplementary Fig. 8). Blue, yellow, and white circles represent glucose, galactose, and hexose, respectively. Red triangles and purple diamonds indicate fucose and sialic acid, respectively. Source data are provided as a Source Data file.

adapted by directed evolution to package and protect their own RNA genome[25].

To decorate these structures with glycans, we introduced glycosylation sites at surface exposed regions of the protein subunits and co-expressed them with ApNGT. We observed an increase in the mass of each of the targeted polypeptides corresponding to the addition of a single hexose (Fig. 3a–c). In the case of the AP205 VLP and AaLS-13, unmodified protein is not detected in the mass spectra, indicating quantitative modification of the proteins. Upon addition of a trivalent tag to AaLS-13, we observed the addition of up to three glucose residues per monomer (average of two hexoses), generating structures that present an average of 720 sugars across the surface (Supplementary Fig. 9). Transmission electron microscopy confirmed that these particle assemblies retain the original morphology upon introduction of glycosylation sites and glycans (Fig. 3).

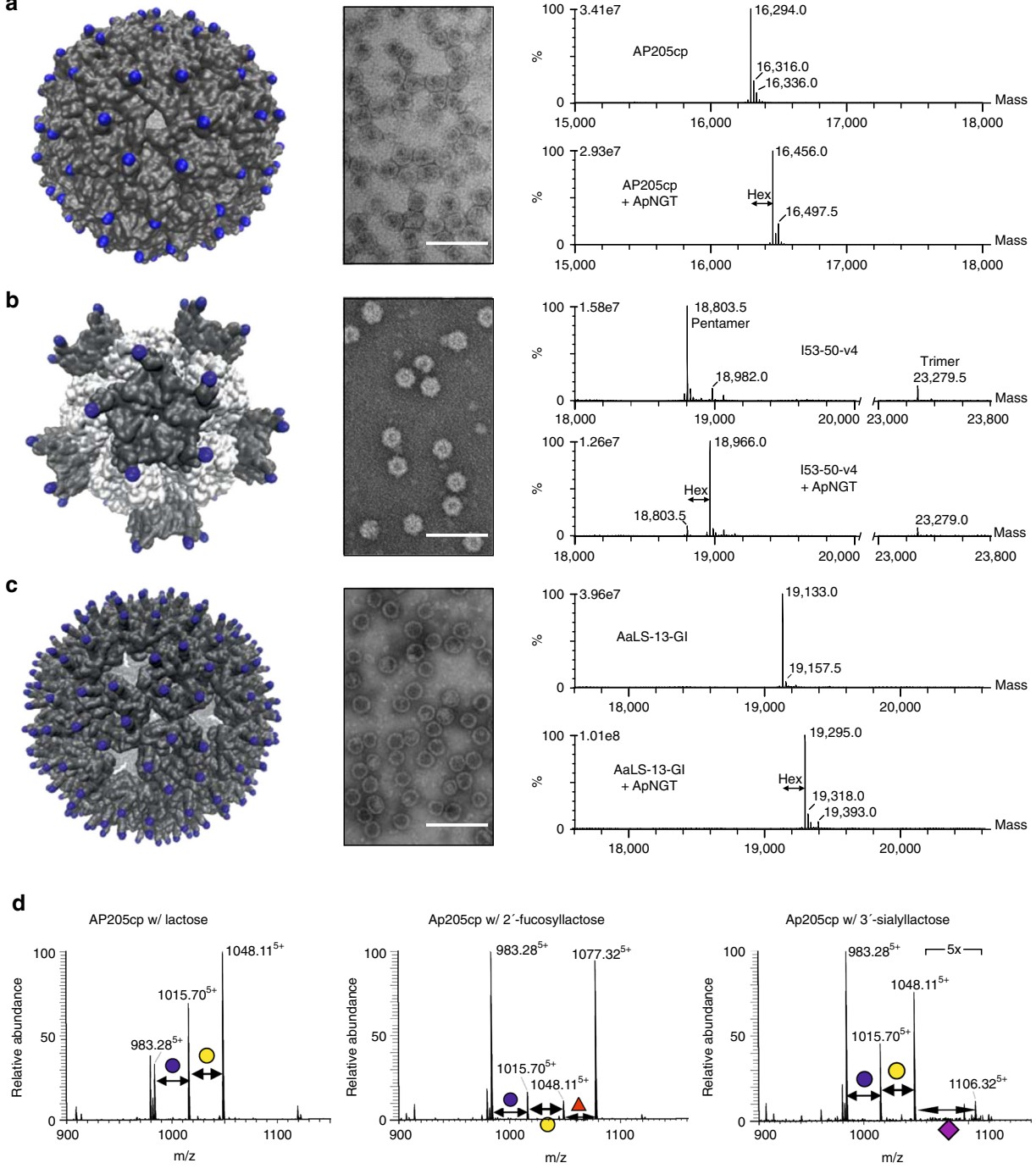

**Fig. 3** Glycosylation of megadalton protein assemblies. Glycosylation sites were introduced into surface-exposed regions of three self-assembling protein nano-structures. Each construct was expressed in the absence and presence of the NGT, particles were purified, and characterized. Glycosylation sites are indicated as blue spheres on the molecular model of each particle; **a** AP205 VLP, **b** I53-50-v4, trimer and pentamer subunits are white and gray, respectively, and **c** AaLS-13. Assembly of particles was assessed by transmission electron microscopy (scale bars represent 100 nm). Glucosylation was assessed by intact protein mass spectrometry, deconvoluted spectra are shown in the right panel. **d** The AP205 VLP was co-expressed with established pathways for lactose, 2′-fucosyllactose, and 3′-sialyllactose. Modification of the coat protein (AP205cp) was assessed by nano-LC–MS/MS analysis of tryptic peptides. MS spectra are the sum of all modified and unmodified peptides, in the 5+ charge state, with sequence TLFASGNAGLGFLDPTAAIVSS DTTAGSGGAHATA**N**ATAHATWSHPQFEK (where **N** is the potential N-glycosylation site). The assigned structures are supported by respective HCD–MS/MS spectra (Supplementary Fig. 10).

In a final step, we expressed the AP205 VLP together with pathways for lactose, 2′-fucosyllactose and 3′-sialyllactose. We were able to confirm the addition of these sugars to the target site on the AP205 coat protein (Fig. 3d and Supplementary Fig. 10). Comparison of the extract ion chromatograms suggest that we have occupancies in the range of 50%, 40%, and 5% for the respective glycans, indicating that some improvement in pathway efficiency will be necessary to achieve quantitative modification of highly multivalent protein assemblies with more complex glycan structures. Native agarose gel electrophoresis confirmed that

particle assembly was not disrupted by these modifications (Supplementary Fig. 10).

Together, these findings demonstrate that cytoplasmic NGT-based glycoengineering is a powerful platform for generation of defined surface-modified proteinaceous assemblies with possible applications as next generation anti-carbohydrate vaccines, drug-delivery vehicles, and multifunction molecular containers.

## Discussion

Chemical synthesis of glycans[51] and glycoproteins[52] is challenging, and synthetic access to many structures requires combined chemo-enzymatic approaches[53,54]. On the other hand, cells and organisms readily produce a variety of complex carbohydrates and glycoconjugates but often only tiny amounts are produced and they are present as components of complex mixtures of similar structures. A practical solution is to genetically engineer cell lines or bacterial strains to produce desired glycoconjugates. To access new areas of glycan and protein structural space, E. coli expression systems have the distinct advantage that they lack native protein glycosylation machinery, thus permitting the design and installation of orthogonal glycosylation pathways.

In this work, we introduce a platform for glycoprotein engineering in E. coli. Two features of this platform make it unique among existing glycoengineering efforts: (i) its use of sequential rather than en bloc glycosylation and (ii) its location in the cytoplasm. Sequential glycosylation enables the construction of a glycan directly on the protein substrate by successive action of different glycosyltransferases. This provides a greatly simplified pathway architecture compared to en bloc glycosylation, which requires preassembly of the glycan as an LLO, enzymatic flipping of the LLO to the lumen of the periplasm, followed by transfer of the entire oligosaccharide onto the protein.

The glycosylation pathways established in this work yield biologically relevant glycan structures, a number of which have important applications as research and medical reagents. 2′-fucose and 3′-sialic acid modifications are present on glycolipids and glycoproteins on the surface of human cells. 3′-sialyllactose is the head group of the GM3 ganglioside and the ligand of avian influenza virus[55]. The incorporation of host-derived GM3 into the membrane of HIV-1 particles mediates infection of immune cells via interaction with Siglec-1[56]. Fucosylated glycans are abundant on the lumenal surface of intestinal epithelial cells and were recently shown to be ligands for cholera toxin[57]. The finding that 2′-fucosyllactose and fucose-bearing polymers are competitive inhibitors of cholera toxin binding[44] suggests promising applications for multivalent fucosylated structures produced with this system.

Two of our prototype pathways address the biosynthesis of capsular polysaccharides from N. meningitidis. The serogroup C polysaccharide is the protective antigen targeted by the meningococcal vaccine incorporated in the current childhood schedule. In order to raise a long-lasting and protective immune response against the polysaccharide, it is first purified from large-scale cultivation of the pathogen, then chemically cross-linked to an immunogenic carrier protein[58]. Recombinant synthesis of complete glycoconjugate vaccines would reduce the cost of production by orders of magnitude, and promises to increase access in developing countries where the burden of disease is highest.

A unique feature of this platform is its location in the bacterial cytoplasm, the most popular compartment for protein expression in engineering and directed evolution studies. Engineered proteins that are cytoplasmically produced include catalysts for treating nerve agent intoxication[59] or leukemia[60], a diversity of antibody mimetics[31], and self-assembling protein nanostructures[18,23–25]. Glycosylation of these proteins has the potential to transform their application as therapeutic reagents by, for example, reducing immunogenicity[61], increasing circulating half-life[32], targeting the molecule to particular cells or tissues[62,63], or inducing a glycan-specific immune response[64]. A caveat of this system is that the initiating enzyme, ApNGT, requires access to the targeting N-x-S/T motif in an unfolded state. In this study, we introduced glycosylation sites on accessible terminal polypeptides and we have previously demonstrated efficient modification of a flexible loop inserted into a folded domain[32]. We expect that glycosylation sites in structured segments of a protein domain are likely to be challenging substrates for this cytoplasmic glycosylation platform.

A further challenge that is addressed in this work is the generation of multivalent glycoconjugates. Due to the relatively low affinities of glycan–protein interactions, biological systems typically exploit multiple simultaneous interactions to achieve functionally relevant avidities[41]. Multivalent interactions are made possible by the dense display of glycan epitopes, for example, the plasma membrane of a mammalian cell is decorated with several million glycans[65], and the mucin, MUC1, can be glycosylated at one in every four amino acids along the polypeptide chain[66]. This platform provides facile access to multivalent glycan display either along a single polypeptide chain or in a defined geometric arrangement on the surface of proteinaceous nanostructures.

For application of multivalent glyco-structures, pathway efficiencies will need to be improved. Although glucose and lactose were efficiently installed on both mono- and multivalent substrates, the assembly of larger oligosaccharides was less efficient. Bottlenecks may result from: (i) insufficient nucleotide sugar availability, (ii) unbalanced expression of glycosyltransferases, and/or (iii) low specific activity of glycosyltransferases towards heterologous substrates. We are currently developing high-throughput screens as a basis for the metabolic and enzyme engineering required to achieve quantitative glycosylation of highly multivalent glycostructures.

In summary, the Glycoli platform advances protein glycoengineering on several fronts. It provides access to mammalian and bacterial glycan structures that are not accessible to existing en bloc glycoengineering platforms. It brings glycoprotein biosynthesis to the cytoplasmic compartment, thus converting the vast majority of engineered proteins—including therapeutic enzymes, antibody mimetics, and self-assembling proteins—into accessible substrates. Finally, it replicates natural forms of multivalent glycan display by enabling the production of heavily glycosylated polymers and particles.

## Methods

**General plasmid design and construction.** Glycosyltransferases used to build-up glycosylation pathways are listed in Supplementary Table 1. Strains and plasmids used in this study are listed in Supplementary Table 2. Each glycosyltransferase BioBrick (GTBb) comprised a single expression cassette including a variable promoter (lacUV5, or T5), a ribosome-binding site, the relevant open-reading frame, and a T7 terminator, all flanked by the BioBrick prefix and suffix (Supplementary Fig. 1). Open-reading frames were codon optimized and synthesized (Genscript), then cloned into the BioBrick expression cassette via BamHI/AvrII. Construction of multi-brick pathways was carried out by standard restriction enzyme-based BioBrick cloning[38]. All constructs were confirmed by Sanger sequencing (Microsynth).

**Construction of plasmids AaLS-13-GI and AaLS-13-GIII.** Plasmid pMG-AaLS-13[20] was used as the starting point for cloning glycosylation competent AaLS constructs. Two DNA fragments, GI-temp and GIII-temp, coding for glycosylation tags containing one and three glycosylation sites, respectively, flanked by a Gly-Ser linker and a C-terminal hexahistidine tag, were generated. GI-temp was constructed from the primers FW_GI_temp (5′-GGATCAGGCGCTCATGCGA CGGCGAACGCTACCGCTCAT) and RV_GI_temp (5′-GTGATGATGGTGA TGGTGGCTAGCATGAGCGGTAGCGTT). GIII-temp was assembled from the primers FW_GIII_temp (5′-GGATCAGGCGCTAATGCGACGGCGAACGCTA

CCGCTAAT) and RV_GIII_temp (5′-GTGATGATGGTGATGGTGGCTAGCA TTAGCGGTAGCGTT). In both cases, overhanging ends were filled in by the Klenow fragment of DNA Polymerase I. XhoI and SpeI sites were introduced using primers FW_XhoI_GI (5′-GCAGCTCGAGGGGGAGTGGATCAGGCGCTCAT GCG) or FW_XhoI_GIII (5′-GCAGCTCGAGGGGGAGTGGATCAGGCGCTAA TGCG) for the 5′ region and RV_SpeI_G (5′-CTCCTCACTAGTTAGTGATGA TGGTGATGGTGGCTAG) for the 3′ region. The resulting 93 bp PCR products were digested with XhoI and SpeI to give 82 bp fragments. These fragments were ligated with the correspondingly digested 4993 bp fragment from pMG-AaLS-13 to obtain plasmids pMG-AaLS-13-GI and pMG-AaLS-13-GIII.

**Bacterial strains and growth conditions**. *E. coli* DH5α was used for maintenance and propagation of plasmids. DH5α cells were cultivated at 37 °C in Luria-Bertani (LB) medium in shaker flasks or on LB plates containing 1.5% agar (w/v), supplemented with appropriate antibiotics (kanamycin 50 μg mL$^{-1}$, chloramphenicol 35 μg mL$^{-1}$, trimethoprim 50 μg mL$^{-1}$).

Expression of glycosylated substrates, both proteins and VLPs was carried out in the *E. coli* K-12 derivative strain W3110 Δ*nan*AK Δ*lacZ*. Knock-out of the *nan*AK prevents catabolism of sialic acid, whilst the β-galactosidase encoding gene *lac*Z was knocked-out to avoid cleavage of N-Lac[36]. Terrific Broth (TB) supplemented with 0.5 M NaCl was used for protein expression. Cultures were grown at 37 °C to an OD600 of ~0.7. Expression was induced by the addition of 1 mM IPTG, and where necessary also 0.4% L-arabinose. Cultures were grown for a further 16–20 h at 28 °C in the presence of inducing agents. In the case of fucosylated structures, the culture was spiked with 2 g L$^{-1}$ of L-fucose. When cultivating strains with pHT081, the culture was spiked with 5 mM Neu5Ac. Cell pellets were harvested by centrifugation, washed once with PBS and stored at −20 °C.

**Purification of GFP constructs**. The cell pellet from a 20 mL expression cultures was resuspended in 5 mL lysis buffer (60 mM Tris pH 8, 1 mM MgCl$_2$, 150 mM NaCl) supplemented with 20 μg mL$^{-1}$ Dnase I and 1 mg mL$^{-1}$ lysozyme (Sigma), Protease Inhibitor Cocktail (Roche), and 2 mM β-mercaptoethanol (Sigma). Cells were lysed by sonication. Samples were cleared by centrifugation at 15,000 × *g*, 15 min at 4 °C. The supernatant was further cleared by filter sterilization prior to affinity purification.

The sfGFP constructs were purified via Ni-NTA beads (Macherey-Nagel). Clarified lysates were loaded onto pre-equilibrated beads, then beads were washed with high salt and urea buffer (60 mM Tris pH 8, 1 M NaCl, 2 M Urea) and washing buffer (60 mM Tris pH 8, 150 mM NaCl, 30 mM Imidazole). Proteins were eluted using elution buffer containing 250 mM Imidazole (60 mM Tris, 150 mM NaCl). Samples were concentrated and buffer exchanged into freezing buffer (50 mM Tris pH 8, 50 mM NaCl, 10% glycerol, 5 mM DTT) using centrifugal filter units with a 10 kDa cut off (Amicon, Millipore).

Protein concentrations were determined by BCA assay (Pierce). Purified (glyco) proteins were separated by Tricine–SDS–PAGE[67]. Gel staining was done using colloidal Coomassie stain[68]. Immunoblots were performed as follows. His tag was detected using mouse anti-His$_4$ antibodies (1:2500, Qiagen, Cat. no. 34670), with goat anti-mouse IgG (1:2500, Scanta Cruz Biotechnology, sc2082). Strep-tagII was detected directly with Streptactin–HRP (1:10000, IBA, Cat. no. 2-1502-001). N-Glc was detected using the human serum, MS14 (1:3500)[28,42], with mouse anti-human IgG Fc (1:5000, Southern Biotech, Cat. no. 9040-05).

**Purification of polysialylated proteins**. The polysialylated fraction of a GFP sample was purified via preparative anion exchange chromatography. The eluate from NiNTA beads was desalted and loaded onto a 2 mL Mono Q column (5/50 GL) pre-equilibrated with buffer A (20 mM Tris pH 7.5). Proteins were eluted in a gradient up to 100% buffer B (20 mM Tris pH 7.5, 1 M NaCl) at a flow rate of 1 mL/min with online absorbance measurement at 280 nm. Polysialylated fractions were collected, desalted, and analyzed by SDS–PAGE and analytical anion exchange chromatography. Polysialylated proteins were further analyzed by immunoblot with antibodies or antisera against the relevant serogroup. To detect proteins carrying the NmB serotype glycan, the anti-PSA IgG2a (clone name: 735), Kappa antibody was used (1:1000, Absolute Antibody, Cat. no. Ab00240-2.0) followed by goat anti-mouse IgG-HRP (1:2500, Santa Cruz Biotech., sc-2005). To detect NmC capsular polysaccharide, antiserum (222301, BD) was purified using a Protein A HP SpinTrap column (GE Healthcare). The eluent was diluted 1:1000 for immunoblotting, then detected with goat anti-Rabbit-IgG-HRP (1:3000, BioRad, 172-1019) secondary. Uncropped images of all blots and gels are provided in the Source Data file.

**Analytical anion exchange chromatography**. For characterizing chain length heterogeneity of polysialylated GFP samples, 3 μg of protein was separated on a ProPac SAX column (Dionex) with online fluorescence detection. Proteins were loaded on a column equilibrated with 20 mM Tris–HCl pH 7.5, then separated with a gradient from 0 to 600 mM NaCl, over 24 min, in 20 mM Tris–HCl at pH7.5.

**Purification of AP205 VLPs**. Expression pellets were resuspended in 5 mL lysis buffer (50 mM Tris pH 6.8, 150 mM NaCl, 1 mM MgCl$_2$, 0.1% Triton x-100) per

gram of wet weight, then supplemented with 1 mg mL$^{-1}$ lysozyme, 20 μg mL$^{-1}$ Dnase I, and 0.5 mM PMSF. Lysis was allowed to proceed for 100 min at 37 °C with shaking. Lysis was completed by sonication. Samples were cleared by centrifugation at 25,000 × *g* during 30 min at 4 °C. AP205 VLPs were purified via StrepTactin Sepharose beads (Macherey-Nagel). VLPs were bound to StrepTactin resin in batch for 60 min at 4 °C. After washing the beads (wash buffer: 50 mM Tris pH 6.8, 150 mM NaCl, 2 mM EDTA), the VLPs were eluted using wash buffer supplemented with 5 mM desthiobiotin (Sigma). Protein concentration was estimated based on absorbance at 260 and 280 nm and accounting for packaged RNA[69]. VLPs were exchanged into freezing buffer (50 mM Tris pH 6.8, 150 mM NaCl, 10% glycerol, 2 mM EDTA) using 100 kDa centrifugal filters (Amicon, Millipore), and stored at −80 °C after freezing in liquid nitrogen.

VLPs were separated by Tricine–SDS–PAGE and native agarose electrophoresis on 0.6% TAE agarose.

**Purification of I53-50-v4**. Expression pellets were lysed as described for the AP205 VLP (above). The I53-50-v4 particles were purified via Ni-NTA beads. Particles were bound in batch, then transferred to a column and washed with 20 column volumes (CV) of wash buffer (50 mM Tris pH 6.8, 150 mM NaCl, 20 mM imidazole). Protein was eluted with four CVs elution buffer (50 mM Tris pH 6.8, 150 mM NaCl, 250 mM imidazole). Samples were concentrated and buffer exchanged into freezing buffer (50 mM Tris pH 6.8, 150 mM NaCl, 10% glycerol) using centrifugal ultrafiltration units (MWCO 100 kDa).

**Production and purification of AaLS-13 constructs**. 14 mL PS tubes (Greiner bio-one 191161) were filled with 5 mL LB media (LB Broth, Miller–DIFCO 244610) containing 100 μg mL$^{-1}$ ampicillin and 25 μg mL$^{-1}$ chloramphenicol. Following inoculation with a single picked colony from BL21-Gold cells transformed with pMA933 and the different AaLS-13 plasmids, the cultures were grown at 37 °C and 230 rpm. When the OD$_{600}$ reached 1.1–1.5, the cultures were induced with 2 μL 1 M IPTG (Fluorochem M02726 to a final concentration of 0.4 mM). After culturing at 25 °C for 18 h and 230 rpm, cells were harvested by centrifugation at 5000 × *g* and 4 °C for 45 min. Cell pellets were stored at −20 °C until use. Each cell pellet was re-suspended in 400 μL 50 mM sodium phosphate buffer, pH 8, containing 300 mM NaCl and 10 mM imidazole and supplemented with lysozyme (0.1 mg mL$^{-1}$), DNase I (5 μg mL$^{-1}$), and RNase A (5 μg mL$^{-1}$) and protease inhibitor cocktail and incubated for 1 h at room temperature before sonication in 1.5 mL Eppendorf tubes. Cellular debris was removed by centrifugation at 22,000 × *g* for 10 min at room temperature. The lysate was loaded onto a 50 μL suspension of a Ni-NTA sepharose resin (Qiagen) in a centrifugal column unit (Spin column 0.1 mL, G-biosciences 786-719) and all centrifugation steps were done at 100 × *g* for 30 s. After washing with 10 CVs of 50 mM phosphate, pH 8, 300 mM NaCl containing 20 and 40 mM imidazole, proteins were eluted with 50 μL 10 mM sodium phosphate, pH 8, containing 100 mM EDTA.

**Transmission electron microscopy**. Nanoparticle suspensions were adsorbed on carbon-coated grids and stained with 2% (w/v) uranyl acetate (pH 4). Grids were examined with an FTS Morgagni 268 electron microscope fitted with a 100 kV tungsten emitter and 1376 × 1032 pixel CCD detector.

**Glycoprotein analysis by LC–MS/MS**. To prepare glycosylated substrates for MS/ MS analysis, 50 μg of protein was taken and incubated for 1 h at 37 °C with 100 μL 50 mM DTT in ABC buffer (0.05 M NH$_4$HCO$_3$ in water, pH 8.5) to alkylate the protein. 100 μL of 65 mM IAA in ABC buffer was added for a further hour at 37 °C. The substrates were digested overnight at 37 °C using either 10 μL of a 1 μg μL$^{-1}$ trypsin in the case of (multivalent) glycosylated proteins or LysC endopeptidase solution for glyco-VLPs. The samples were acidified the next morning using formic acid, prior to ZipTip (Millipore) clean-up of the samples.

Samples were analyzed on a calibrated Q Exactive$^{TM}$ mass spectrometer (Thermo Fisher Scientific) coupled to a Waters Acquity UPLC M-Class system with a Picoview$^{TM}$ nanospray source 500 model (New Objective). The tryptic samples were dissolved in 2% acetonitrile, 0.1% formic acid, loaded onto an Acclaim PepMap 100 trap column (75 μm × 20 mm, 100 Å, 3 μm particle size) and separated on a nanoACQUITY UPLC BEH130 C18 column (75 μm × 150 mm, 130 Å, 1.7 μm particle size), at a constant flow rate of 300 nL min$^{-1}$, with a column temperature of 50 °C and a linear gradient of 2−22% acetonitrile, 0.1% formic acid in 59 min, and then 22−32% acetonitrile, 0.1% formic acid in 11 min, followed by a sharp increase to 98% acetonitrile in 2 min and then held isocratically for another 10 min. For DDA analysis, one scan cycle comprised of a full scan MS survey spectrum, followed by up to 12 sequential HCD scans based on the intensity. For glycosylation profiling analysis, full-scan MS spectra (300–1800*m/z*) were acquired in the FT-Orbitrap at a resolution of 70,000 at 400*m/z*, while HCD MS/MS spectra were recorded in the FT-Orbitrap at a resolution of 35,000 at 400*m/z*. HCD MS/MS spectra were performed with a target value of 1e5 by the collision energy setup at a normalized collision energy of 22. To confirm each glycopeptide structure, MS and MS/MS spectra of the corresponding species were annotated manually.

**Reporting summary**. Further information on research design is available in the Nature Research Reporting Summary linked to this article.

## Data availability

The authors declare that most data supporting the findings of this study are available within the paper, Supplementary Information, and the Source Data file. The raw data underlying peptide LC–MS/MS analysis will be made available by the corresponding author upon request. The source data underlying Figs. 1b–f, 2a, b, and Supplementary Figs. 4a–c, 7e, and 10a, b are provided as a Source Data file. Any other relevant data is available from the authors upon request.

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

## Acknowledgements

We thank Mathilde Brambati and Regina Men Men Wong for technical assistance. We thank Professor Anna Maria Papini for the generous gift of MS14 serum. We are grateful to LimmaTech Biologics AG for the generous gift of *E. coli* strain W3110 *nan*AK *lac*Z. The authors thank the Functional Genomics Center Zurich for providing mass spectrometric analysis and support in interpreting the data. We are grateful to ScopeM for support with transmission electron microscopy. H.L.P.T. acknowledges the European Union Horizon 2020 research and innovation program for her Marie Sklodowska-Curie fellowship (Glycoli, No. 703577). M.B.T. was supported by the Austrian Science Fund (FWF): [J4230-B32]. T.G.K. was supported by an ETHZ postdoctoral fellowship. This work was supported by grant number 310030_162636 to M.A. from the Swiss National Science Foundation, and by Career Seed Grant number SEED-33 16-1 awarded to T.G.K. from the Swiss Federal Institute of Technology. We also thank the European Research Council (ERC) Advanced Grant ERC-AdG-2012-321295 and ERC-2017-PoC-CfC (to D.H.). N.T. and Y.A. were supported by a Human Frontier Science Program Long-Term Fellowship and an ETH Zurich Postdoctoral Fellowship, respectively.

## Author contributions

H.L.P.T. designed and constructed the glycosylation pathways, carried out protein expression, purification, chromatographic and electrophoretic analyses, analyzed results, and drafted the manuscript. C.-W.L. carried out and interpreted peptide mass spectrometric analysis of glycoprotein structure. M.B.T. carried out protein expression, purification, and quantitative chromatographic analyses of protein glycosylation. C.R. developed methods for purification and analysis of polysialylated proteins. J.M. and N.L. assisted with glycoprotein production and characterization. M.D.L., N.T. and Y.A. designed and carried out experiments with AaLS-13. D.H., M.W. and M.F.B. contributed to design of the study. T.G.K. and M.A. designed the study. T.G.K. designed glycosylation pathways and wrote the manuscript.

## Competing interests

The authors declare no competing interests.
