## [Peer Review File · Nature Communications]

Reviewers' Comments:

Reviewer #1:

Remarks to the Author:

This paper describes the assembly and characterization of glycosylation pathways engineered into the cytosol of bacterial that provide a novel form of in vivo glycosylation for recombinant products. The present studies are an extension of prior work by the Aebi lab following the identification of a novel soluble cytoplasmic N-glycosyltransferase, ApNGT, that acts as a to initiate glycosylation at acceptor sequons that are analogous to those on mammalian glycoproteins. This enzyme transfers a glucose residue to form an amide linkage with Asn acceptors that can then be extended by co-expression of other engineered glycosyltransferases in the bacterial cytosol. The present study demonstrates the utility of the in vivo glycosylation system for extension of glycans to form a variety of structures as well as glycan addition to a variety of acceptor proteins including three distinct megadalton self-assembling protein complexes.

The manuscript is exceptionally well-written, provides a strong Introduction for setting the groundwork for the rationale and justification for the respective studies, and then continues to provide numerous examples for applications of the glycosylation technology (termed Glycoli platform). Co-expression of ApNGT with a collection of additional glycosyltransferases led to the formation of extended sialylated or fucosylated structures on GFP reporter proteins with single or multiple acceptor sites. The authors then extended the model systems for cytosolic glycosylation to include a set of 3 self-assembling proteins, AP205cp, I53-50-v4, and AaLS-13-GI. In each case a demonstration of glycosylated product was observed and retention of self-assembly was maintained. The result appears to be a reasonably robust demonstration of efficacy for an in vivo glycosylation machinery that is capable of modifying a diverse collection of recombinant products based on the co-expression of ApNGT for initiation of protein modification with a single core Glc residue onto the polypeptide backbone.

While the general presentation of the platform is well described, there are a few caveats that are not discussed or are not adequately emphasized in the present manuscript.

1) The authors indicate that many of the modification reactions do not go to completion and many result in glycosylated products with far less than full stoichiometry. In addition, some of the reactions were not well characterized in terms of site occupancy. On page 10 the authors indicate that higher expression of ApNGT led to greater occupancy of the resulting glycosylation sites and referred to Fig. 2b. This is not what appears to be presented in Fig. 2b. It would be preferable for this data be explicitly presented in a more controlled manner to test the factors that contribute to glycosylation efficiency. Is initial glycosylation directly dependent on expression level of ApNGT (ratio of ApNGT to glycosylation substrate)? Can the other reactions (or at least one other reaction) be driven to higher occupancy with higher relative expression? Will this be a limitation of the glycosylation machinery in *E. coli*? Is there an indication that modulating GT expression levels will lead to different levels of occupancy?

2) While the respective glycosylation sites that were inserted into the various reporter proteins were often effectively glycosylated, in almost each case the glycosylation site was associated with a disordered N- or C-terminal peptide tail segment. Does the ApNGT effectively glycosylate sites that are internal to a folded protein domain? Does the enzyme only work on terminal accessible tails or internal highly mobile loops? It is important for the authors to at least comment on potential limitations of the glycosylation systems for protein modification. The narrative implies that all designed sites could be modified, but does not address the fact that the design parameters for glycosylation site placement were highly skewed toward accessible terminal peptide segments. A greater discussion of all limitations of the platform would provide a more informed presentation of the utility of the in vivo pathway engineering approach.

Overall, the manuscript is a superb demonstration of a highly versatile platform for glycosylation pathway engineering. Proof of concept studies were presented with the generation of a variety of glycan structures and a collection of glycosylated substrates. While the description is convincing,

the manuscript would be more effective if a clear demonstration of improved site occupancy could be shown through titrating expression level of one or more GT as an indication that further engineering is possible. In addition, a greater discussion of the potential limitations of the platform should be included so the readers could understand how the glycosylation machinery could best be deployed and where challenges may limit applications for generating glycosylated products. If these minor concerns are addressed, there is no question that the manuscript will be a landmark advancement in the glycosylation field that will extend the utility of generating defined glycoproteins on recombinant products for numerous critical applications.

Reviewer #2:

Remarks to the Author:

Dr. Keys and coworkers describe in this manuscript an alternative approach to the periplasmic glycosylation, which has been the mainstay in the field. The cytoplasmic sequential glycosylation based on ApNGT, or Glycoli, as demonstrated in this work, represents a significant advance for the improvement of bacterial protein glycosylation. The many varied glyco-structures assembled onto different protein substrates, some in a multivalent fashion, suggests that this approach of glycosylation is quite promising.

A few suggestions for the authors to consider in revision:

1. P4, line 58, "the availability of a gene cluster" is not sufficient for the glycan to be synthesized.
2. p4, "enticing", exciting?
3. p6, last sentence, not having been demonstrated with other platforms?
4. p8, line 35: Provide the percentage of glycosylated proteins

This is important as one of the disadvantage of the periplasmic approach is the low percentage of glycoprotein.

5. Provide information on potential heterogeneity of the glycans, glycans ending with different sugars

6. p9: what are the yield of glycoproteins with polysaccharide chains, occupancy rate, glycan microheterogeneity?

7. p10. Does glycosylation efficiency depend on the number of repeats?

8. p10: last paragraph. Provide quantitative data.

9. fig. 2a, two upper left panels, shouldn't TAGANATA band be higher than that of GANATA? Large repeats seem also dimmer. Lower expression?

10. p13, yield info compare to single glycosylation site?

11. p14, low glycosylation percentage Any ideas where might be the bottleneck?

Dear Reviewers,

Thank you for your interest and thoughtful comments on the manuscript. A common theme was the inclination to see more quantitative data (yields, occupancies, and distribution of glycoforms). In response, we have carried out considerable additional MS and chromatographic analyses of intact glycoproteins. In several cases, we are now able to provide the average quantities of all glycoforms present on a particular glycoprotein product. The additional analyses support our original findings and generally strengthen the estimated occupancies provided in the original manuscript (those were based on peptide MS analyses). We are satisfied that the requested revisions have pushed us to provide a much improved manuscript and look forward to your further comments.

Below are our point-by-point responses to the concerns raised. Modifications to the text of the revised manuscript are highlighted in blue. Please note that Figure 1 has also been almost completely revised using newly acquired data to provide a quantitative view of the respective glycoproteins and their heterogeneity.

Sincerely,

Tim Keys
On behalf of the authors

Reviewer #1 (Remarks to the Author):

This paper describes the assembly and characterization of glycosylation pathways engineered into the cytosol of bacterial that provide a novel form of in vivo glycosylation for recombinant products. The present studies are an extension of prior work by the Aebi lab following the identification of a novel soluble cytoplasmic N-glycosyltransferase, ApNGT, that acts as a to initiate glycosylation at acceptor sequons that are analogous to those on mammalian glycoproteins. This enzyme transfers a glucose residue to form an amide linkage with Asn acceptors that can then be extended by co-expression of other engineered glycosyltransferases in the bacterial cytosol. The present study demonstrates the utility of the in vivo glycosylation system for extension of glycans to form a variety of structures as well as glycan addition to a variety of acceptor proteins including three distinct megadalton self-assembling protein complexes.

The manuscript is exceptionally well-written, provides a strong Introduction for setting the groundwork for the rationale and justification for the respective studies, and then continues to provide numerous examples for applications of the glycosylation technology (termed Glycoli platform). Co-expression of ApNGT with a collection of additional glycosyltransferases led to the formation of extended sialylated or fucosylated structures on GFP reporter proteins with single or multiple acceptor sites. The authors then extended the model systems for cytosolic glycosylation to include a set of 3 self-assembling proteins, AP205cp, I53-50-v4, and AaLS-13-GI. In each case a demonstration of glycosylated product was observed and retention of self-assembly was maintained. The result appears to be a reasonably robust demonstration of efficacy for an in vivo glycosylation machinery that is capable of modifying a diverse collection of recombinant products based on the co-expression of ApNGT for initiation of protein modification with a single core Glc residue onto the polypeptide backbone.

While the general presentation of the platform is well described, there are a few caveats that are not discussed or are not adequately emphasized in the present manuscript.

1) The authors indicate that many of the modification reactions do not go to completion and many result in glycosylated products with far less than full stoichiometry. In addition, some of the reactions were not well characterized in terms of site occupancy.

The reviewer correctly points out that site occupancy and glycan heterogeneity were not always quantitatively characterized in the original manuscript. These parameters cannot be reliably established from the peptide-LC-MS analysis that we used to confirm glycan structures and site of attachment in the original manuscript. In the revised manuscript, we use intact protein mass analysis to provide a robust, semi-quantitative characterization of these parameters for a number of glycosylation pathways. These include modification of GFP at a single site with:

glucose

lactose

3'-sialyllactose and

2'-fucosyllactose

These new data are incorporated into the manuscript on page 7, lines 127-135, as well as in a revised version of Figure 1. Further, the distribution of glycoforms is also displayed as a stacked histogram in Supplementary Figure 5.

On page 10 the authors indicate that higher expression of ApNGT led to greater occupancy of the resulting glycosylation sites and referred to Fig. 2b. This is not what appears to be presented in Fig. 2b. It would be preferable for this data be explicitly presented in a more controlled manner to test the factors that contribute to glycosylation efficiency. Is initial glycosylation directly dependent on expression level of ApNGT (ratio of ApNGT to glycosylation substrate)?

True, this was not well described in the original manuscript. We have now carried out additional intact protein MS analyses to provide a quantitative comparison of products from the two ApNGT expression vectors. The experiment is described in detail in the revised manuscript (see page 9, lines 184-193). The mean percentage of completely modified protein (with 5 glucoses) from both ApNGT expression vectors is reported in the text.

Can the other reactions (or at least one other reaction) be driven to higher occupancy with higher relative expression? Will this be a limitation of the glycosylation machinery in *E. coli*? Is there an indication that modulating GT expression levels will lead to different levels of occupancy?

We do not have an example of a reaction (except the ApNGT) being driven to higher occupancy with higher relative expression levels. We do have an example of a GT (LgtA) where the product was only observed when the GT was placed behind a stronger promoter. We highlight this explicitly in the revised manuscript (see page 7, lines 135-141). Regarding limitations of glycosylation machinery more generally, we identify three (interrelated) factors that may limit synthesis of the target sugar: i) insufficient GT expression, ii) low GT activity towards (unnatural) glycoprotein substrates, and iii) nucleotide sugar availability. We highlight these factors and a mitigation strategy in the discussion (page 15, line 305-312).

2) While the respective glycosylation sites that were inserted into the various reporter proteins were often effectively glycosylated, in almost each case the glycosylation site was associated with a disordered N- or C-terminal peptide tail segment. Does the ApNGT effectively glycosylate sites that are internal to a folded protein domain?

The ApNGT can glycosylate internal sites in a folded domain, but generally with only very low occupancy.

Does the enzyme only work on terminal accessible tails or internal highly mobile loops?

Practically, yes, for efficient glycosylation it is necessary that the glycosite is present on a flexible polypeptide tag or loop.

It is important for the authors to at least comment on potential limitations of the glycosylation systems for protein modification. The narrative implies that all designed sites could be modified, but does not address the fact that the design parameters for glycosylation site placement were highly skewed toward accessible terminal peptide segments. A greater discussion of all limitations of the platform would provide a more informed presentation of the utility of the in vivo pathway engineering approach.

We agree completely and now provide a greater discussion of limitations of the system w.r.t. the site of glycosylation on substrate proteins (page 14, line 290-295).

Overall, the manuscript is a superb demonstration of a highly versatile platform for glycosylation pathway engineering. Proof of concept studies were presented with the generation of a variety of glycan structures and a collection of glycosylated substrates. While the description is convincing, the manuscript would be more effective if a clear demonstration of improved site occupancy could be shown through titrating expression level of one or more GT as an indication that further engineering is possible. In addition, a greater discussion of the potential limitations of the platform should be included so the readers could understand how the glycosylation machinery could best be deployed and where challenges may limit applications for generating glycosylated products. If these minor concerns are addressed, there is no question that the manuscript will be a landmark advancement in the glycosylation field that will extend the utility of generating defined glycoproteins on recombinant products for numerous critical applications.

Reviewer #2 (Remarks to the Author):

Dr. Keys and coworkers describe in this manuscript an alternative approach to the periplasmic glycosylation, which has been the mainstay in the field. The cytoplasmic sequential glycosylation based on ApNGT, or Glycoli, as demonstrated in this work, represents a significant advance for the improvement of bacterial protein glycosylation. The many varied glyco-structures assembled onto different protein substrates, some in a multivalent fashion, suggests that this approach of glycosylation is quite promising.

A few suggestions for the authors to consider in revision:

1. P4, line 58, "the availability of a gene cluster" is not sufficient for the glycan to be synthesized.

We have corrected the text to be more accurate. It now states that a "functional biosynthetic pathway for producing the LLO" is the requirement (page 4, line 58).

2. p4, "enticing", exciting?

We have replaced "enticing" with "exciting" (page 4, line 78).

3. p6, last sentence, not having been demonstrated with other platforms?

We use the more accurate formulation provided by the reviewer (page 6, line 111).

4. p8, line 35: Provide the percentage of glycosylated proteins. This is important as one of the disadvantages of the periplasmic approach is the low percentage of glycoprotein.

The point that the original manuscript lacked quantitative characterization of sugar occupancy and heterogeneity is well taken. We have carried out extensive intact protein mass analysis to provide a comprehensive, albeit semi-quantitative, characterization of occupancy and heterogeneity for a set of glycoproteins that are central to the manuscript. These include GFP modified at a single site with:

glucose

lactose

3'-sialyllactose and

2'-fucosyllactose

In addition, the analysis of 3'-sialyllactose modified GFP was backed up with analytical anion exchange chromatography (in good agreement with the numbers from intact protein MS).

These new data are incorporated into the manuscript on page 7, lines 127-135, as well as in a revised version of Figure 1. The distribution of glycoforms is displayed as a stacked histogram in Supplementary Figure 5. Analytical anion exchange analysis of 3'-sialyllactose modified GFP is displayed in Supplementary Figure 6.

We also provide quantification of GFP-[GANATA]₅ modified with glucose (page 9, line 189-193), and lactose (page 10, line 200-201).

5. Provide information on potential heterogeneity of the glycans, glycans ending with different sugars

Please see the response to point 4 (above). The distribution of glycoforms for a set of products is explicitly displayed in Supplementary figure 5.

6. p9: what are the yield of glycoproteins with polysaccharide chains, occupancy rate, glycan microheterogeneity?

To provide this data, we have produced and purified the polysialylated proteins and quantified yield, occupancy and heterogeneity of the glycoproteins. The data is now included in the manuscript (page 8, lines 158-165) and in a revised version of Figure 1 (see panel f and g).

7. p10. Does glycosylation efficiency depend on the number of repeats?

We can address this question with the newly acquired quantitative data on glucose addition to the single- and five-site GFP substrate proteins. There does appear to be a simple relationship: more sites places a greater demand on the glycosylation pathway/enzymes leading to reduced efficiencies. For example, although glucosylation proceeds to 100% on a single-site acceptor (page 7, line 129-131), using the same ApNGT expression plasmid, it only proceeds to 93% on the five-site "[GANATA]₅" acceptor (page 10, line 189-191).

8. p10: last paragraph. Provide quantitative data.

We have now carried out additional intact protein MS analyses to provide a quantitative comparison of the NGT expression vectors. The experiment is described in detail in the revised manuscript (see

page 9, lines 184-193) and the mean percentage of completely modified protein (with 5 glucoses) from both NGT expression vectors is noted in the text.

9. fig. 2a, two upper left panels, shouldn't TAGANATA band be higher than that of GANATA? Large repeats seem also dimmer. Lower expression?

We have not observed significant differences in expression levels between the different GFP expression runs. Slight differences in band size (and to some extent running behavior) is likely due to small differences in the amount of protein loaded. All samples displayed in 2a were run on the same gel and an image of the uncut gel is provided in the source data file - so it may be fully scrutinized. In our experience, these results are consistent with expectations given the resolution of SDS-PAGE. We note that the same samples were analyzed by peptide-LC-MS/MS analysis providing a chemical confirmation of the sequence (Supplementary Figure 8).

10. p13, yield info compare to single glycosylation site?

The yield compared to single glycosylation sites is reduced. In the revised manuscript we include quantification (based on intact protein mass analysis) of protein fully occupied with lactose at five-sites (69%). We were unable to accurately quantify occupancy of the 2FL and 3SL structures due to lower occupancies and the large number of intermediate structural variants. However, the peptide-LC-MS/MS analysis suggests the portion of fully modified proteins is low (<5%). We include this information in the revised manuscript (page 10, line 200-204). Possible bottlenecks are discussed (page 15, line 305-312).

11. p14, low glycosylation percentage Any ideas where might be the bottleneck?

AP205cp is a bacteriophage protein that is expressed at extremely high levels (>250 mg/L). This high burden of protein synthesis competes directly with production of the glycosylation enzymes. Further, it results in a large number of glycosylation sites in the cytoplasm, placing a higher burden on the glycosylation pathway than, for example, the single-site GFP construct. These factors together likely cause the observed reduced glycosylation efficiency. We include a discussion of the generic factors that are likely to limit glycosylation efficiency in the revised manuscript (page 15, line 305-312).

Reviewers' Comments:

Reviewer #1:

Remarks to the Author:

This is a revised submission of a paper describing the assembly and characterization of glycosylation pathways engineered into the cytosol of bacterial that provide a novel form of in vivo glycosylation for recombinant products. The critique of the prior submission expressed concerns about the stoichiometry of the glycosylation reactions, appropriate quantitation of the respective reaction products, and issues related to the positions of the glycosylation site insertions. The revised manuscript addresses all of these concerns both through the inclusion of additional data (intact MS of glycosylated products) and expansion of the discussion in the Results and Discussion regarding the limitations of the platform. It now provides an effective presentation of the utility of the in vivo pathway engineering approach.

Overall, the revised manuscript was very responsive to the prior concerns and represents a superb demonstration of a highly versatile platform for glycosylation pathway engineering. The manuscript will be a landmark advancement in the glycosylation field that will extend the utility of generating defined glycoproteins on recombinant products for numerous critical applications. It is acceptable for publication in its present form.

Reviewer #2:

Remarks to the Author:

The revision provided several quantitative details of the technology, which are very helpful for the reviewer to assess the challenges of this glycosylation approach. Overall, the authors did a great job in addressing the various questions raised by this reviewer.

One minor revision is suggested on the description of LacNAc and Lewis X antigen synthesis, p7, lines 135-139. The sentences were awkward.

Also, given low percentage glycosylations for some glycans, it might be more appropriate to emphasize novel glycans accessible with this technology. P6, the end of the second paragraph, the authors could list explicitly what these novel structures are.

Dear Reviewers,

We are grateful for the time you have invested reviewing this manuscript and for your constructive comments. Our responses to requests for minor revisions are detailed below (in blue).

Kind regards,

Tim Keys

REVIEWERS' COMMENTS:

Reviewer #1 (Remarks to the Author):

This is a revised submission of a paper describing the assembly and characterization of glycosylation pathways engineered into the cytosol of bacterial that provide a novel form of in vivo glycosylation for recombinant products. The critique of the prior submission expressed concerns about the stoichiometry of the glycosylation reactions, appropriate quantitation of the respective reaction products, and issues related to the positions of the glycosylation site insertions. The revised manuscript addresses all of these concerns both through the inclusion of additional data (intact MS of glycosylated products) and expansion of the discussion in the Results and Discussion regarding the limitations of the platform. It now provides an effective presentation of the utility of the in vivo pathway engineering approach.

Overall, the revised manuscript was very responsive to the prior concerns and represents a superb demonstration of a highly versatile platform for glycosylation pathway engineering. The manuscript will be a landmark advancement in the glycosylation field that will extend the utility of generating defined glycoproteins on recombinant products for numerous critical applications. It is acceptable for publication in its present form.

Reviewer #2 (Remarks to the Author):

The revision provided several quantitative details of the technology, which are very helpful for the reviewer to assess the challenges of this glycosylation approach. Overall, the authors did a great job in addressing the various questions raised by this reviewer.

One minor revision is suggested on the description of LacNAc and Lewis X antigen synthesis, p7, lines 135-139. The sentences were awkward.

These sentences have been revised and simplified in the final version of the manuscript.

Also, given low percentage glycosylations for some glycans, it might be more appropriate to emphasize novel glycans accessible with this technology. P6, the end of the second paragraph, the authors could list explicitly what these novel structures are.

We have re-phrased this sentence to emphasize that all nine N-glycans described in this manuscript are only accessible with this technology.